# Data Reliability and Trustworthiness through Digital Transmission Contracts

Simon Mangel[1], ✉ Lars Gleim[1], Jan Pennekamp[2],
Klaus Wehrle[2], and Stefan Decker[1,3]

[1] Databases and Information Systems, RWTH Aachen University, Germany
[2] Communication and Distributed Systems, RWTH Aachen University, Germany
[3] Fraunhofer FIT, Sankt Augustin, Germany

**Abstract.** As decision-making is increasingly data-driven, *trustworthiness and reliability* of the underlying data, e.g., maintained in knowledge graphs or on the Web, are essential requirements for their usability in the industry. However, neither traditional solutions, such as paper-based data curation processes, nor state-of-the-art approaches, such as distributed ledger technologies, adequately scale to the complex requirements and high throughput of continuously evolving industrial data. Motivated by a practical use case with high demands towards data trustworthiness and reliability, we identify the need for *digitally-verifiable data immutability* as a still insufficiently addressed dimension of data quality. Based on our discussion of shortcomings in related work, we thus propose *ReShare*, our novel concept of digital transmission contracts with bilateral signatures, to address this open issue for both RDF knowledge graphs and arbitrary data on the Web. Our quantitative evaluation of ReShare's performance and scalability reveals only moderate computation and communication overhead, indicating significant potential for cost-reductions compared to today's approaches. By cleverly integrating digital transmission contracts with existing Web-based information systems, ReShare provides a promising foundation for data sharing and reuse in Industry 4.0 and beyond, enabling digital accountability through easily-adoptable digitally-verifiable data immutability and non-repudiation.

**Keywords:** Digital transmission contracts · Trust · Data immutability · Non-repudiation · Accountability · Data dynamics · Linked Data · Knowledge graphs

## 1   Introduction

With current trends in the Internet of Things and Industry 4.0, decision-makers nowadays have access to a wide variety of data sources and platforms, thus enabling a data-driven decision-making process. Developments in Open Data underline the trend towards open data sharing paradigms independent of the domain in question. E.g., supply chains already demonstrate these trends in productive use [3,16], where novel approaches enable multi-hop data sharing [30], effectively forming an Internet of Production (IoP) [29] where multiple stakeholders collaborate. Here, benefits include reductions in costs, increased profit margins, and general improvements in production quality [29].

To realize novel use cases in an IoP, data trustworthiness and reliability are crucial properties to enable sound data-driven decisions [25,26,35]. Neglecting these aspects

can cause severe damages, such as miscalculations, economic losses, or even harmful to humans [2,10,24,28]. Apart from trust and reliability, *interoperability* is important for any data sharing [13]. In this matter, Linked Data (LD) [1], a paradigm for inter-business data sharing, is a promising candidate. Additionally, LD also facilitates provenance tracing [15] and thus can positively influence trust. Overall, assessing trust and reliability in the context of LD is extremely relevant.

A promising approach to establish objective trustworthiness is to enrich data with digital signatures [40], as automatic signature creation and verification promise scalable solutions. Related work [7,20,36] proposes several existing approaches regarding the signing of Linked Data, which serves as a foundation for any approach using signatures of Resource Description Framework (RDF) resources. However, the rising needs w.r.t. scalability in the face of increasing data dynamics are rarely considered. Hence, existing approaches towards signing LD nowadays have limited applicability. That is, while coarse signatures often force users to retrieve unnecessary data only to verify signatures, fine-granular signatures, which offer more utility by design, impact the scalability due to the communication and storage overhead caused by a large number of signatures, causing a trade-off which impairs the achievable scalability. Moreover, signatures independently generated by the data source cannot provide immutability, as the data source can simply forge signatures for modified data. The reliability guarantees of existing signature-based approaches are significantly weaker than Distributed Ledger-based approaches, where strong immutability is created by committing the state of data to an immutable ledger [16,34]. However, these systems suffer from limited scalability due to limited throughput and infrastructure overhead [38].

To address the required goals of trustworthiness and reliability through scalable immutability, we propose an on-demand signature scheme, where the sender and receiver actively engage in the transmission process and both sign a so-called **D**igital **T**ransmission **C**ontract (DTC). In contrast to common signature approaches, DTCs establish the *immutability* of data reliably, as both peers of a transmission would have to collude to forge a DTC. Together with the non-repudiation provided by the signatures, immutability further implicates *accountability*, which is relevant in the face of liability conflicts [30]. Thus, DTCs enable transmission peers to exchange trustworthy and reliable data by creating immutability based on bilateral signatures with improved scalability w.r.t. data dynamics through their on-demand methodology. Our mechanism is designed to be flexible towards arbitrary data formats, such as conventional, unlinked data, as well as Linked Data resources by employing canonicalization.

**Contributions.** Our main contributions in this paper are as follows.

– We coin the need for data immutability as an enabler of trustworthiness and reliability, unlocking new data sharing and collaboration use cases based on well-founded data-driven decision making.
– Addressing today's shortcomings, our novel design of digital transmission contracts allows users to verify and prove immutability besides the integrity and authenticity of exchanged data at a low overhead in terms of both computation and communication.
– We further provide the research community with a detailed assessment of the applicability and feasibility of our approach ReShare and thereby create the foundation for the novel paradigm of on-demand bilateral signatures for scalable immutability.

## 2    Design Goals

To address the lack of suitable approaches in reliable and trustworthy data sharing, we identify a set of concise design goals, which will guide us through the review of related work and presentation and evaluation of our approach.

**G1: Integrity, Authenticity & Non-repudiation.**  The ability to digitally verify the integrity and authenticity of LD resources is essential to establish objective trustworthiness. Thereby, unauthorized modification is prevented, and the users' trust in data correctness is strengthened. Further, non-repudiation is needed for accountability (cf. **G2**), as the possibility to repudiate a given action hinders a party's accountability for said action. As all three requirements can be fulfilled using digital signatures, we group these three desired properties under a single goal.

**G2: Immutability.** To strengthen data reliability and to establish accountability, we postulate that any used system must be made immutable. Third parties should not be able to easily question this immutability. That is, the system should provide proofs of immutability, which malicious entities cannot easily forge.

**G3: Applicability.**  To be viable in realistic use cases, the solution should be directly applicable and provide both trustworthiness and reliability. Interoperability and usability further affect the applicability of a system, as they facilitate integration in use cases.

**G4: Scalability.** Given that the increasing data dynamics, especially in the industrial domain, are a significant challenge for data consumers, any proposed signing and verification approach must scale to future needs, i.e., it should be able to timely react to the frequent creation and modification of data without significantly impairing the design.

**G5: Performance.** To complement **G4**, we emphasize that unreasonable overhead for any involved party must be strictly avoided. Otherwise, the proposed solution limits their ability to participate and leads to undesired constraints following a restricted throughput and consequently a non-acceptance of the system in real-world settings.

**G6: Payload Flexibility.** As a scalable approach towards trustworthy and reliable data sharing (cf. **G1**, **G2**) unlocks important use cases in industrial environments where the LD paradigm is not yet well-established, we argue that a sole focus on LD significantly limits the applicability. Thus, we demand flexibility concerning the payload of the created signatures. For improved adaptability, any proposed system should support arbitrary data formats, with a specific focus on commonly employed formats on the Web.

Note that the entirety of our design goals exceeds the needs for applications that are exclusive to the Semantic Web. Indeed, any approach that fulfills **G1**-**G5** can satisfy the requirements of data trustworthiness and reliability in the Semantic Web. However, we intend to propose a more flexible and all-encompassing industry-ready approach.

## 3    Background & Related Work

Now, we outline related work and fundamental concepts for the challenge of data trustworthiness and reliability. First, we relate the general problem of data quality to our goals of trustworthiness and reliability, before we summarize existing approaches to signing LD resources. With the issue of data mutability in mind, we briefly discuss distributed ledger technology that promises data immutability. Throughout the section, we discuss the shortcomings of approaches based on our design goals described in Sec. 2.

**Data Quality and Trust in the Semantic Web.** Data quality is commonly conceived as its *fitness for use* w.r.t. a given application [40]. Thus, it constitutes a heterogeneous concept with dimensions that partly are of subjective or context-dependent nature. Zaveri et al. [40] were able to extensively identify and categorize sub-dimensions and metrics of data quality in the context of LD.

As one of the six categories of data quality dimensions, *trust* severely suffers from more dynamic associations of stakeholders, such as prevalent in modern supply chains with increasing flexibility [3,27]. Zaveri et al. [40] investigate trust in detail and identify *reputation*, *believability*, *verifiability*, and *objectivity* as dimensions of trust. The metrics of reputation, believability, and objectivity are either only able to indicate trustworthiness, e.g., by checking for the existence of meta-information about the data source, or depend on a sophisticated trust model, which may not exist in real-world use cases.

Consequently, we believe that none of these three dimensions allows for objectively assessing the trustworthiness of data independent from its context when the fourth dimension – verifiability – is not given, as fraudulent modification and forgery are not prevented. Contrarily, verifiability can be objectively assessed by the use of digital signatures. As long as the signature is bound to the data source, e.g., by employing a Public-Key Infrastructure (PKI) [31], signatures grant authenticity, integrity, and non-repudiation of the data. We argue that data trustworthiness may be sufficiently asserted through signature verification if the data source is trusted.

**Approaches towards LD Signatures.** After discussing the concept of data quality in the Semantic Web, we now survey existing approaches that sign LD resources. Note that the discussed approaches fulfill **G1**, i.e., the ability to sign LD resources, by design.

To the best of our knowledge, Carroll et al. [7] were the first to propose a sophisticated signature mechanism specifically for usage with Linked Data. The authors especially focus on the canonicalization algorithm and argue that, even if graph canonicalization is GI-complete, practically graphs can be canonicalized in $\mathcal{O}(n \log(n))$. In this regard, Carroll et al. [7] are relevant for any task which needs a canonical RDF representation. However, the authors focus on the signature mechanism itself and do not propose a complete system, as the use of a PKI and the distribution of signatures are not discussed.

Tummarello et al. [36] followed up on Carroll et al. [7] by proposing a more holistic system for LD signatures. The authors argue that graph-level signatures [7] are often too coarse for practical use cases, as users would always have to request the entire RDF graph to verify the signature. To address this issue, the authors proposed to sign the data at a much finer level, i.e., at the level of Minimum Self-contained Graphs (MSGs). Signatures are attached to an arbitrary triple of the signed MSG, thus internalizing them. By directly attaching signatures and certificate metadata to the data itself, the authors explicitly specify how to apply the system to a knowledge graph (**G3**). However, as also criticized by Kasten et al. [20], the certificate is referenced by a URI, which makes the signature unusable if a certificate can no longer be retrieved. Furthermore, a user has to compute at least a partial partitioning into MSGs to know which statements were signed in a given signature, as such information is not explicitly stated. Moreover, the approach has severe issues concerning scalability (**G4**) and performance (**G5**) caused by the fine granularity of signatures. If no blank nodes are present, each statement is signed separately, resulting in a substantial overhead caused by the signatures, which is even

exaggerated if data is modified frequently. Due to the focus on LD, the approach does not support signing other data formats, i.e., it does not provide payload flexibility (**G6**).

Kasten et al. [20] improved on Tummarello et al. [36] by proposing a framework that allows for signatures at different levels of granularity, i.e., reaching from MSG signatures up to signing multiple graphs at once. The authors discuss and formalize the entire signature process in a framework and only give exemplary solutions for the identified functions. Therefore, the work does not constitute a directly applicable solution, but rather aims at building a foundation for applicable solutions through formalization. Due to its improved flexibility, the approach provides better scalability (**G4**) and reduced overhead (**G5**) compared to Tummarello et al. [36]. **G6** is not met, as the approach specifically focuses on RDF data. Most importantly, all approaches listed above have a common crucial deficiency w.r.t. our design goals, i.e., they do not create immutability (**G2**). As the data source only signs data, this entity can easily forge signatures for modified data at any time, thus violating immutability. This aspect is crucial, as we identified immutability as a core requirement for data reliability in Sec. 2.

To the best of our knowledge, none of the existing approaches to signing LD resources can sufficiently fulfill our design goals, especially w.r.t. immutability (**G2**).

**Distributed Ledger-based Immutability.** In contrast to signatures, Distributed Ledger (DL) technology is designed to provide strong immutability (**G2**) by committing the state of data to an irreversible ledger [41]. This property is highly desirable and thus celebrated in a wide variety of use cases, such as distributed supply chains [3,16]. Such use cases with little pre-existing trust relations and opportunities to employ LD technology motivate the combination of LD and DL technology to establish immutability. Consequently, use cases and barriers w.r.t. the combination of LD and DLs have been identified [6,12,33]. Furthermore, researchers proposed the first concrete solutions employing DL technology to establish immutability in the context of LD [34].

However, the intersection of the two research fields still is in its infancy, and existing approaches rarely cover the need for immutability holistically. Furthermore, the strong immutability provided by Distributed Ledgers comes with practical disadvantages. Even with new consensus mechanisms such as the Swirlds hashgraph consensus algorithm [4] or the tangle [32], which try to mitigate the negative effect of common, costly consensus mechanisms such as proof-of-work, DL systems always bring a substantial overhead (**G5**). This overhead is infeasible in many use cases, as with huge amounts of data, e.g., produced by IoT sensors, the relative cost for conducting the consensus mechanism exceeds the value of the data written to the ledger. Thus, we decided to refrain from involving DL technology in our system. However, we think that the intersection of Distributed Ledgers and LD in scenarios where LD immutability is crucial and the imposed overhead is acceptable makes for a promising research area for future work.

In conclusion, we found that none of the existing approaches towards LD signatures were able to sufficiently fulfill our design goals, especially due to the missing immutability (**G2**). Research regarding combinations of DL and LD technology to provide strong immutability (**G2**) is still in its infancy, resulting in prohibitive scalability (**G4**) and performance (**G5**) overhead for many use cases as detailed in our supplementary material [18]. Therefore, we identify the requirement for an approach that bridges the needs for immutability, scalability, and performance.

## 4   ReShare: Reliable Data Sharing through DTCs

To address the previously identified shortcomings of related work w.r.t. our design goals, we propose *ReShare*, a scalable and flexible on-demand resource signature system. ReShare employs the novel concept of **D**igital **T**ransmission **C**ontracts (DTCs). Whenever the state of data needs to be proven as immutable, the data sender and receiver partake in a *contract generation handshake*, which results in the creation of a DTC. A DTC is a record comprising the state of the subject data (as checksums), the identities of sender and receiver, as well as a timestamp of contract creation, crafted immutably by adding signatures of *both* peers. In the DTC generation mechanism, the receiver requests a DTC for a set of resources. The sender compiles said DTC by creating checksums for the resources, adding metadata, and signing the record. Finally, the receiver signs the retrieved DTC and sends its signature back to the sender.

The validity of a contract can be automatically verified at any time in a corresponding *contract verification mechanism*, where the identities of the parties, the signatures, and the resource checksums are verified. We expect that creating signatures for data exchanges offers improved scalability (**G4**) compared to existing signature mechanisms in use cases where the amount of produced and modified data outweighs the number of data requests. This expectation is strengthened by ReShare's ability to bundle multiple resources into one DTC, thus reducing the per-resource signature overhead (**G5**). Furthermore, we argue that ReShare provides immutability in addition to the existing benefits of digital signatures, i.e., integrity and authenticity, as a colluding of two parties is needed to forge a valid contract that violates the data immutability.

First, we proceed by motivating a use case in the domain of aerospace engineering. Subsequently, we present details about both the contract generation and verification mechanisms. Finally, we discuss realization aspects of ReShare, i.e., the representation and ontology of DTCs and the integration of ReShare with LD technology.

**ReShare's Capabilities Illustrated using Aerospace Engineering.**  For our system, aerospace engineering constitutes an interesting use case, as reliability is highly desirable when designing and producing safety-critical products. This need even is legally justified, as federal US-American laws require manufacturers to securely store the *type design*, comprising drawings and specifications, information about dimensions, materials, and processes for as long as the respective *type certificate* of the aircraft is valid (cf. 14 CFR §21.31, §21.41, §21.49 [37]). Thus, such data must be kept available reliably, at least as long as an aircraft of the given type is operational. As a result, manufacturers usually apply specialized archiving systems [22]. However, if we consider modern IoT-backed supply chains, massive amounts of data can hardly be processed by common archiving pipelines, which may still involve humans in paper-based signature mechanisms [39]. Therefore, aerospace engineering is a prime candidate to integrate ReShare.

To further look into this use case, we illustrate the benefits of employing an LD-based approach for tracking and tracing through a practical example: Involving the manufacturer Boing, which conducts the final assembly of an aircraft, the independent supplier ACom, which provides the radio unit for Boing's aircraft together with relevant production data, and regulatory agency FÄA that ensures legal compliance. We further consider the following scenarios to visualize the system's functionality:

**Assembly.** Boing assembles an aircraft using a radio unit supplied by ACom, while ACom further grants Boing access to all relevant production data. Enriched with metdata, such as provenance information, this data is stored in an LD platform. Boing and ACom also generate a DTC, which is bound to the state of said data. In this context, both Boing and ACom verify their certificates and signatures, thus mutually establishing authenticity. Due to the dataflow direction, we refer to ACom as the *sender* and to Boing as the *receiver*.

**Proof of Conformance.** After an aircraft was involved in an incident, Boing has to prove to FÃA that certain requirements were met during manufacturing. To this end, Boing presents the relevant data, together with the respective DTCs. The contracts include all relevant context to verify the authenticity and state of the data. If an investigation of the incident concludes that, for example, the data regarding the radio is incorrect, Boing is not liable, as it acted to the best of its knowledge. Rather, ACom can be held accountable.

**Tracking and Tracing.** To deal with the aftermath of a defective radio unit, Boing traces all other aircraft that use the same type of radio using the data's semantic properties. While an improved tracking & tracing efficiency is not a contribution of our system, we argue that ReShare provides the needed reliability through LD.

After outlining three distinct and common real-world scenarios, we now present ReShare's DTC generation and verification mechanisms.

### 4.1 Generation Mechanism

The contract generation mechanism constitutes a relatively simple 3-way handshake, which we visualize in Fig. 1. We use this opportunity to address the different types of identifiers used in the system. Resources are identified by unique Resource IDs (RIDs). While ReShare supports arbitrary RIDs, the use of Internationalized Resource Identifiers [11] (IRIs) improves interoperability with LD technology. For identification of the contract itself, ReShare mandates the use of unique IRIs [11] in order to integrate DTCs with common LD technology. The mechanism works as following, where **R** denotes the receiver, and **S** the sender:

R1: **R** chooses a set of RIDs and sends the RIDs and his identity, including a public key certificate, to **S**.

S1: **S** can now assemble the contract associating checksums of the canonicalized resources identified by the RIDs. Then, **S** creates a timestamp and a unique contract IRI and adds them to the contract. Finally, **S** creates its signature with the JSON representation of the contract as input. Then, the signature is added to the contract. **S** then sends the assembled contract, including its identity with a public key certificate, its signature, the RIDs with associated checksums, the timestamp, and the contract IRI back to **R**.

R2: **S**'s signature (i.e., `senderSig`) is verified, together with **S**'s certificate (included in the `sender` field). The timestamp recency is checked. **R** signs the contract similarly to **S**.

The complete contract, including both peers' signatures, is finally sent to **S**.

Both signatures are created by omitting any existing signatures in the contract as an input to the signature creation. That is, both the sender and receiver sign the DTC, including the identities, resource checksums, and the timestamp. The computation overhead on the sender's side for creating the contract, which mainly consists of generating checksums

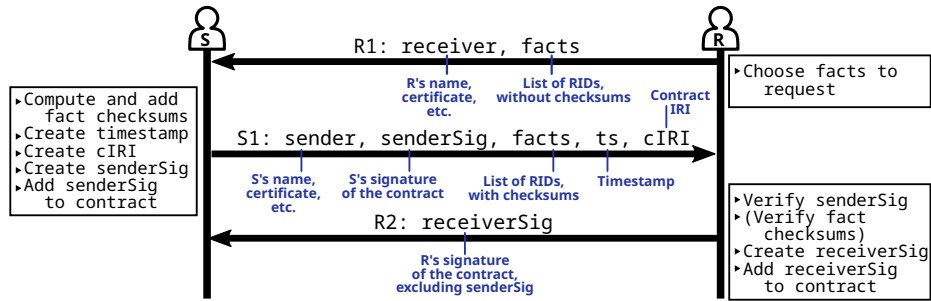

**Fig. 1.** Visualization of the contract generation handshake. **S** is the *sender*, **R** is the *receiver*. Here, *Fact* is used as a synonym for a persistently identified resource.

and creating the signature, can be reduced using pre-generated checksums in use cases where scalability needs are especially high, or Denial of Service by R1 flooding is a valid threat model. In the latter case, the problem could otherwise also be mitigated by the use of rate limiting or access control.

The messages contain complete, incremental versions of the contract, which allows the sender to remain stateless. This design fits our server-client model nicely, where the sender as a server provides an interface for receivers to request contracts. The handshake is always executed on top of TCP, which guarantees reliable communication. Therefore, an error indicates a faulty or incompatible configuration. A mechanism to deal with errors is not part of ReShare, but could be easily implemented for future work.

### 4.2   Verification Mechanism

Contract verification relies only on access to the contract and the covered data itself and consists of the following steps (in no particular order):

- – Verify the public key certificates using the respective PKI
- – Verify the signatures using the JSON-formatted contract and the public keys
- – Verify the data checksums

Hence, only access to the contract and the data are needed. As ReShare currently only supports a CA-based PKI with X.509 certificates, the necessary PKI context comprises a set of pre-installed root CA certificates to verify the X.509 certificate chains of both parties. Contracts are explicit about all information needed to verify signatures and checksums, e.g., about the canonicalization and serialization of the data. To verify the checksums, all parties use the same input as during the DTC generation, i.e., the full DTC excluding any existing signatures. Through the simplicity of the verification, only consisting of the three steps described above, the system is applicable (**G3**) in use cases with needs for unambiguous verifiability of the generated DTCs, e.g., in legal conflicts.

### 4.3   Realization

We implemented ReShare as a Proof-of-Concept Node.js module [18]. We discuss our decisions concerning the implementation and used technologies in the following.

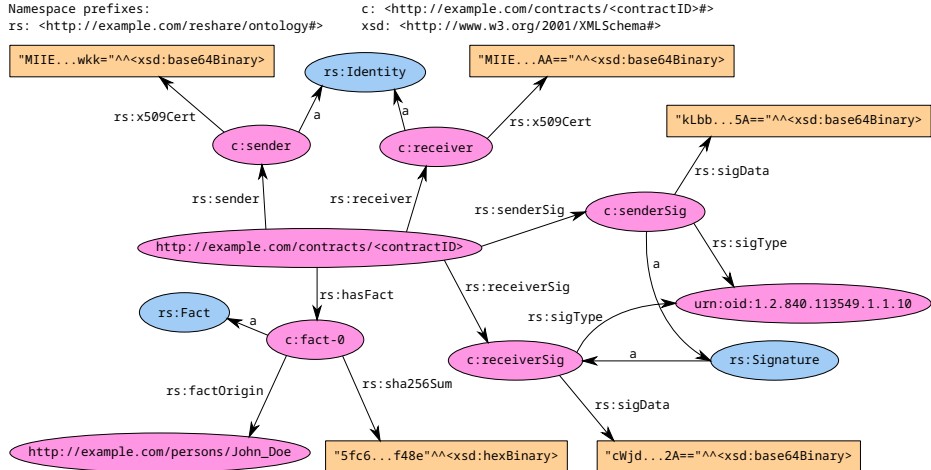

**Fig. 2.** Structure of an exemplary ReShare contract, visualized as an RDF graph using Turtle notation. `http://example.com/contracts/<contractID>` here symbolizes the contract IRI chosen by the sender (cf. Sec. 4.1).

**Contract Ontology and Representations.** As Tummarello et al. [36] discuss, the ability to internalize signatures into the context of the signed data improves the overall usability, because data and signatures are directly associated. Therefore, contracts are represented as JSON-LD [21] by default. For this, we have defined the ReShare ontology [19], which defines the types and properties in a contract. This makes contracts themselves usable in LD platforms. A DTC contains the following:

– A root node, identifying the contract itself, identified transparently and uniquely by the contract IRI chosen by the sender
– A set of resource checksums, associated with their resources by RIDs (cf. Sec. 4.1)
– The identity of sender and receiver, including X.509 certificates [8]
– The signatures of sender and receiver (`RSASSA-PSS` [23])
– A timestamp in the ISO 8601 format (interpretable as `xsd:dateTimeStamp` [9])

An example contract RDF graph with the default structure can be seen in Fig. 2. Note that the notion of *Facts* originates from the FactDAG data interoperability model [13] and its implementation FactStack [14], where a revisioning system is used to create persistence. As this paradigm is not mandatory in ReShare, we use *fact* as a synonym for any persistently identified resource in this paper.

Because we require payload flexibility (**G6**), DTCs should also be compatible with other common technology stacks outside of the Semantic Web. Therefore, in addition to the JSON-LD representation, the context can be omitted, resulting in a pure JSON representation of DTCs, making contracts usable as structured data.

**Contract Generation.** Given that the contracts are represented as JSON-LD or JSON by default, we decided to rely on a JSON-based protocol for contract generation, where the JSON contracts are wrapped into a minimalistic message structure. The JSON-LD context is automatically added when exporting the contract after generation.

The most basic generation mode corresponds to an execution of the JSON protocol directly on top of TCP, with optional use of TLS. However, this approach requires opening a dedicated port for ReShare. Therefore, we also provide an HTTP(S) mode to integrate ReShare into existing Web servers. Then, the 3-way handshake is wrapped into two HTTP POST requests. Thus, we end up with four modes, which we denote by TCP (i.e., without TLS or HTTP), TLS (i.e., TCP+TLS without HTTP), HTTP, and HTTPS.

## 5 Evaluation

To assess the benefits, possible limitations, and applicability of ReShare, we first quantitatively evaluate the storage and communication overhead as well as the effects of latency to the generation mechanism, before qualitatively discussing the fulfillment of our design goals as defined in Sec. 2, whilst giving outlooks to promising use cases.

### 5.1 Quantitative Evaluation

To quantitatively evaluate the performance of the system, we simulated contract generation with varying (i) modes of operation (TCP/TLS/HTTP/HTTPS), (ii) number of facts per contract, and (iii) certificate chain lengths of both peers. We quantitatively evaluate the (a) total duration of the handshake, (b) the number of bytes transferred, as well as (c) the size of the generated contract in its default JSON representation. We employ a TCP proxy to investigate the impact of varying network latency on the protocol's performance and measure the amount of data transferred. Overhead for TLS and HTTP are included in the results. We split the evaluation into two orthogonal parameter combinations to facilitate visualization and discussion, which we show in Table 1.

**Contract Size & Communication Overhead.** In Fig. 3, we show how the number of facts in one contract influences contract bytes and communication overhead per fact, split by handshake mode. A per-fact plot brings better comparability to other approaches than per-contract, as contracts are a concept that is specific to our approach. Thus, metrics are plotted on a per-fact (i.e., per-resource) basis.

The total contract size and communication overhead per handshake increase linearly with the number of facts, as fact data is of constant size, consisting of a RID and a checksum. Thus, if $m$ models one of the per-contract metrics, this gives $m(n) = sn + c$, where $s$ is the slope of the curve, i.e., the bytes by which the metric increases if one fact is added per contract, $n$ is the number of facts per contract, and $c$ is the constant overhead which is not influenced by the number of facts. Then, we can model the per-fact metric

**Table 1.** Overview of the evaluated parameter combinations.

| Protocol Mode | Facts/DTC | Proxy delay [ms] | Cert. Chain Len. | Iterations |
|---|---|---|---|---|
| 1 TCP, TLS, HTTP, HTTPS | 1, 5, 10, 20, 30, 40, 50, 60, 70, 80, 90, 100 | 0 | 1, 2, 3, 4, 5 | 20 |
| 2 TCP, TLS, HTTP, HTTPS | 10 | 0, 10, 20, 30, 40, 50, 60, 70, 80, 90, 100 | 1 | 50 |

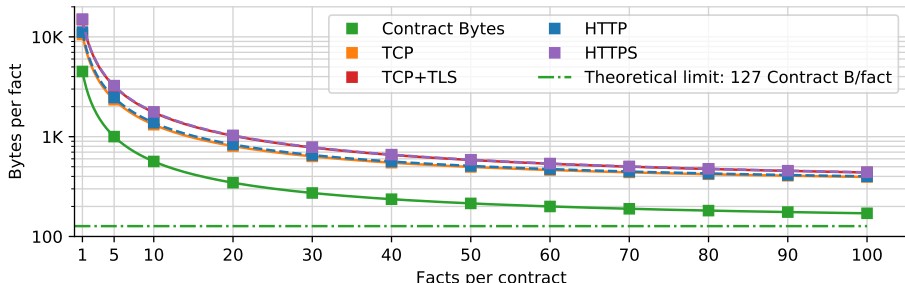

**Fig. 3.** Contract bytes and communication bytes per fact, by the number of facts in one contract. Dataset 1 from Table 1 was used. Communication bytes are split up by handshake mode, i.e., HTTP enabled/disabled, and TLS enabled/disabled. Compared to a per-contract plot, this representation provides better interpretability when comparing the system to other approaches, as contracts are an unknown concept in other work.

as $m'(n) = \frac{m(n)}{n} = s + cn^{-1}$. This model is plotted for each metric. The theoretical limit for contract bytes per fact naturally is given by $s$, which is the slope of the linear per-contract fit. It can be interpreted as the number of bytes that are caused by a fact itself, excluding the static overhead in contracts, which is independent of the number of facts. An analog interpretation of the other metrics slopes is possible. The Figure also shows that the overhead of HTTP is negligible, whereas enabling TLS causes a constant communication overhead of approximately 200 B per contract.

**Handshake Duration.** To better evaluate the handshake duration in a realistic scenario, we simulate varying degrees of network latency. The results can be seen in Fig. 4. Because the relationship between network latency and total handshake duration expectedly is linear, the slope of a linear fit divided by 2 gives a rough estimate on the number of Round Trip Times (RTTs) a contract generation takes.

As a first observation, our implementation produces a constant overhead of approximately 140 ms in handshake duration. Similar to the previously evaluated communication overhead in bytes, HTTP and TCP add the least latency overhead. As the three handshake messages (cf. Sec. 4.1) can directly be sent on top of TCP or HTTP without additional messages, the slope comes close to the baseline of a 3-way handshake. If TLS is enabled, the latency effect is significantly increased by the TLS handshake, i.e., for each TLS handshake, the delay increases by approximately 1 RTT. In TLS mode without HTTP, one socket is used for all messages. Thus, only one TLS handshake is needed, causing a 1 RTT overhead compared to raw TCP. As our implementation currently does not support socket reuse when using HTTP, we need two TLS handshakes in HTTPS mode (one for each POST request), causing a 2 RTT overhead compared to HTTP without TLS, resulting in a maximum handshake duration of approximately 3.5 RTT. Note that reusing cryptographic material from the first request does not reduce latency overhead, as it does not change the number of TLS handshake messages. sages.

For future work, we plan to add socket reuse support for HTTP to our implementation.

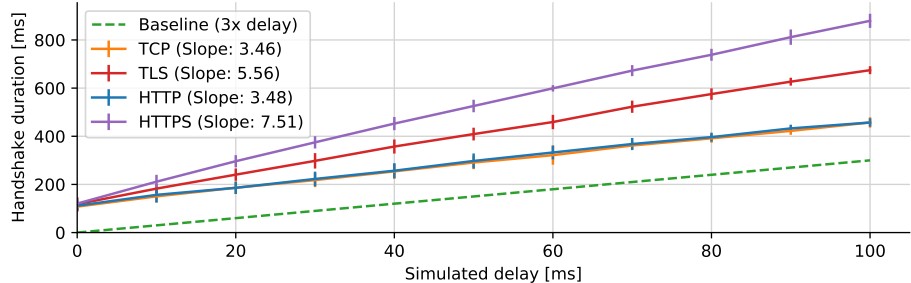

**Fig. 4.** Visualization of the effect of communication delay in-between sender and receiver simulated with an artificial delay in the TCP proxy. Dataset 2 from Table 1 was used. The data was collected using 10 facts per contract and a certificate chain length of 1. Per mode and artificial delay, 50 contracts were generated. The error bars display the interval of $2\sigma$, thus accounting for approximately 95 % of the measurements. The baseline represents a 3-way handshake without any overhead by computation or additional messages.

## 5.2   Qualitative Evaluation

After quantitatively evaluating the performance of ReShare, we now classify the quantitative results and discuss the benefits and disadvantages of ReShare w.r.t. the design goals defined in Sec. 2, as well as practical considerations.

**Trustworthiness & Reliability.** In Sec. 2, we identified signatures as a core enabler of data reliability and trustworthiness (**G1**), which are extremely relevant aspects of data quality (cf. Sec. 1). For the created signatures to create trustworthiness, the entire trust chain, reaching from the PKI as the trust root data checksums, has to be validated as defined in Sec. 4.2. Because the verification algorithms, i.e., X.509 certificate validation, RSA-PSS signature verification, and SHA-2 checksum validation, are generally accepted, our focus is on the availability of the necessary signatures, checksums, and certificates.

First, the material necessary for verifying the signatures themselves is contained within the DTC. Second, for the included certificate to be verifiable, the consumer has to trust the root CA which signed the peer certificates. This assumption is reasonable, as root stores and root programs have long-established extensive CA curation [17]. However, the expiry and revocation of X.509 certificates, as well as the retrievability of the root certificate, which is not included in the contract, hinder verifiability, and thus, reliability (**G1**) in application scenarios with long-term storage requirements. One could counteract this with special approaches for long-term signature preservation, such as Bralić et al. [5], which have detrimental effects on performance (**G5**) or scalability **G4** and require additional infrastructure. We leave this issue to be investigated in future work. Third, the data has to be available in order to verify the included checksum. Here, the structure of DTCs has the advantage that not all included data has to be available in order to keep the signature material verifiable, as individual checksums can be verified. Thus, if certain resources are no longer needed, they can simply be deleted without impacting the verifiability of other resources signed by a contract.

Overall, DTCs are easily verifiable, thus providing good reliability and trustworthiness (**G1**), as long as the certificates are valid and the root certificate is retrievable.

**Immutability & Accountability.** Usually, data-driven business models with high data reliability needs either depend upon well-trusted business partners or have to resort to both time- and cost-intensive manual data curation [39]. Application of ReShare thus provides promising opportunities to create trustworthiness where trust cannot easily be established otherwise and can further be used as a legal binding of the peers to the underlying transmission, useful in legal conflicts. ReShare provides improved immutability compared to usual signature schemes, as illegally forging a valid DTC requires collusion of both peers of the contract. Because benign behavior of the two peers is a severely limiting assumption, ReShare cannot hold up with DL-based immutability, as successful Distributed Ledgers are considered to be irreversible, unless a large share of the network colludes. Thus, ReShare provides enhanced immutability (**G2**) compared to other signature-based approaches, but cannot keep up with DL technology regarding this aspect. To improve the immutability guarantees of ReShare, one could employ a digital notary, e.g., by adding additional signatures by impartial third parties or committing contracts to a DL. However, to keep scalability, one should incorporate measures to reduce the notary overhead, e.g., by only using the notary for interval-based checksums of all created contracts. We deem this idea an interesting direction for future work.

**Performance & Scalability.** If a sufficiently large number of resources are signed per-contract, the per-resource storage communication overhead falls below 1 kB relatively quickly. With a handshake latency overhead of less than 200 ms (without delay), less than 5 RTTs, and the simple contract verification mechanism, we argue that the amount of imposed overhead by the use of the system is reasonably low, especially in comparison to traditional proof of transmission approaches, such as paper-based receipts commonly used in industry today [39]. Therefore, the performance goal (**G5**) is met. One could argue that if individual transmissions only consist of a few resources (e.g., only a few RDF statements), the per-resource overhead both for storage of the contracts and the handshake increases relatively fast. This issue also exists in related work w.r.t. LD signatures, as the severity of the overhead imposed to the user when using coarse signatures is exaggerated in this scenario (cf. Sec. 3). However, if, on the one hand, the overall frequency of requests is low, this issue becomes less severe, as the throughput requirements are small. If, on the other hand, higher request frequencies are expected, ReShare provides the opportunity of resource bundling, i.e., requested resources can simply be buffered by the client and bundled into a single DTC, thus mitigating the issue. In environments with high request frequency by many distinct data recipients, DTC bundling may, however, only apply to a lesser degree. However, besides a limitation w.r.t. these specific circumstances, ReShare provides decent scalability (**G4**).

**Payload Flexibility.** Because DTCs use checksums of canonicalized data, the data format is arbitrary, as long as a canonical representation is specified, which contributes to payload flexibility (**G6**) and allows for applicability to generic Semantic Web data and any other type of resource on the Web. Thus, ReShare constitutes a unified solution for arbitrary data on the Web.

**Other Practical Considerations.** ReShare has the advantage that it is optionally adaptable both for individual stakeholders and individual transmission, as its use is not

mandatory. If one installs ReShare, but opts out to generate contracts for transmissions, made possible through the optionally adaptable design, the system generates little to no overhead. Suppose that it is used in HTTP(S) mode, then, it can be integrated into an already existing web server, and thus does not require additional hardware, infrastructure, or specific software. Such a deployment is useful where manual data curation may be more cost- or time-efficient than generating DTCs, or peers simply do not implement ReShare, making the system fully backward-compatible.

The peer X.509 certificates make up for a majority of the contract data. Thus, removing the certificate data from DTCs and instead of referencing peer certificates with unique identifiers would drastically reduce contract sizes, improving scalability (**G4**) and performance (**G5**). However, as verifiability is the key to the provided trustworthiness and reliability, we argue that making the verifiability of DTCs dependent upon the certificate availability would substantially weaken verifiability, and thus, our core requirement (**G1**). However, to practically reduce the overhead imposed by peer certificates, one could assign unique identifiers to the used certificates in the LD context of DTCs, which allows to only store the certificates once when using an LD platform such as a triple store. This method could be realized with ReShare as part of future work.

ReShare can also be integrated with existing systems and data, i.e., it is backward-compatible. Retrospectively generating DTCs even has an advantage w.r.t. performance (**G5**) and scalability (**G4**), as all resources from a given data source can be bundled into a single DTC, thus reducing the per-resource communication and storage overhead. However, using retrospectively generated contracts, one can not prove possession of the data for the time interval before contract generation due to the contract timestamp.

To conclude, with decent scalability in most use cases and stronger immutability than common signature schemes, we see no significant limitations for ReShare's applicability, making it a promising solution for a variety of use cases with requirements of scalability, trustworthiness, and reliability.

## 6    Conclusion

In this paper, we expressed the need for immutability as an enabler for data trustworthiness and reliability, paving the way for novel use cases employing LD technology for reliable data sharing and collaboration. After identifying the lack of suitable solutions that bridge the need for scalability and immutability, we present ReShare, our design utilizing digital transmission contracts to establish immutability through signatures by both transmission peers, imposing a reasonably low overhead with good scalability. We provide the research community with a discussion of feasibility and applicability, building a foundation for future work w.r.t. scalable immutability for real-world use.

Trustworthiness and reliability are essential requirements for a more open data sharing paradigm in the industry, as economic outcomes depend on data correctness. Digital signatures can provide integrity, authenticity, and non-repudiation, and therefore, they can be used to create data trustworthiness. However, we argue that simple signatures cannot reliably establish immutability, as the signing authority can forge arbitrary signatures, thus hindering data reliability. Recently, distributed ledgers are frequently proposed to achieve the immutability of information. Unfortunately, their scalability is substantially

challenged through limited throughput and infrastructure overhead. To address these issues, we propose ReShare, a system for creating on-demand bilateral signatures contained in digital transmission contracts. Given that both transmission peers would have to collude to forge valid digital transmission contracts, we argue that ReShare provides improved immutability compared to common signature systems, combined with proper scalability through moderate overhead and the ability to sign multiple resources at once.

In our evaluation, we demonstrate that our proposed design shows promising applicability, as its immutability is valuable in use cases with high scalability needs, while being flexible towards the format of signed data and optionally adaptable by concept, imposing little overhead when peers opt-out from usage. Thus, ReShare is a prime candidate to achieve data reliability and trustworthiness for both the Semantic Web and industry. For future work, optionally-adaptable notary systems could further strengthen the immutability of our proposed contracts. However, already in its current state, ReShare allows for novel approaches that profit from and build upon the proposed concept of on-demand bilateral signatures.

**Acknowledgments**  Funded by the Deutsche Forschungsgemeinschaft (DFG, German Research Foundation) under Germany's Excellence Strategy – EXC-2023 Internet of Production – 390621612.

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
