# OpenReview forum: "Data Reliability and Trustworthiness through Digital Transmission Contracts"
_eswc-conferences.org/ESWC/2021/Conference/Research_Track — ESWC 2021 Research_

### Official Review · AnonReviewer3 · 2021-01-11
**Review of article "Data Reliability and Trustworthiness through Digital Transmission Contracts"**

**Confidence:** 4
**Impact:** 2
**Design And Technical Quality:** 2

**Review:**

The authors propose ReShare, a protocol to address the need for "digitally-verifiable data immutability".
The article is well written (only some minor language problems). The topic is interesting and important, but I am not sure that the contribution is significant.

Pros:
- The authors clearly identify design goals (section 2)
- The authors make an effort to design a protocol that generalizes to non linked-data formats
- The authors use standard techniques (digital signatures, X509 certificates, JSON-LD, etc.)

Cons:
- No verification that the proposed protocol is flawless
- Some parts of the protocol are not fully explained and therefore are not convincing
- The example (section 4.1) is not convincing
- The quantitative evaluation (section 5.1) does not show scalability (which is a main design goal)

**Anonymity:**

Yes, I would like my review to remain anonymous.

**Rating:**

-1: Weak Reject

**Reuse And Availability:**

3: Medium

**Strong Points:**

- The authors clearly identify design goals (section 2)
- The authors make an effort to design a protocol that generalizes to non linked-data formats
- The authors use standard techniques (digital signatures, X509 certificates, JSON-LD, etc.)

**Subreviewer:**

I submitted this review.

**Weak Points:**

- Section 4.1: "type design" data may be "massive", but I think it does not change often (high volume but low velocity). Therefore there is no need to frequently recompute signatures, and traditional well-established approaches can handle requirements. In other words, I think that "type design" data does not require scalability (dynamics and velocity) as described in your design goal G4, and therefore the example is weak.
- Section 4.1: "However, if we consider modern supply chains, employing an IoT paradigm, massive amounts of production and design data can hardly be processed by common archiving pipelines": define massive.
- Section 4.1: general comment: it is a bit difficult to follow the example without knowing how DTC works. Perhaps it is more clear if you move the example after the section explaining DTCs.
- Section 4.2: "R chooses a set of RIDs": this is one of my main concerns in the proposed protocol: how does R know in advance the RIDs? And if R already knows the RIDs why asking them again from S (the DTC as illustrated in figure 1 will just send back the RIDs with their checksums from S to R)?
- Figure 1: have you done any (formal) analysis to show that the proposed protocol is safe and flawless?
- Figure 1: unclear: in the data exchanges you use "etc." (for example "R's name, certificate, etc."): you need to be precise.
- Figure 1: what does "S's signature" cover? Everything including the timestamp?
- Figure 1: language: you use both "S's signature" and "senderSig": I assume they are the same, but I suggest to use only one term
- Figure 1: S "computes and add fact checksums": if R requests millions of facts, computing their checksums may take a considerable amount of time: how do you handle this case, especially with respect to your design goal G5?
- Figure 2: I think there should be also a timestamp, which is listed as a part of DTC in section 4.4
- Section 4.4: "the contract IRI chosen by the sender": there is an assumption here that the sender will maintain the contract IRI indefinitely. Is this reasonable?
- Table 1: I am not quite sure what is the real difference between the settings in line and line 2
- Section 5: I think you should have evaluations with (a) large numbers of facts (for example 1 million, 10 millions, etc.), and (b) with facts changing frequently (IoT use case). (a) and (b) would be useful to prove scalability
- Section 5.1: "Handshake Duration": "the slope of a linear fit divided by 2 gives a rough estimate on the number of Round Trip Times": why?
- Section 5.2: "Thus, if certain resources are no longer needed, they can simply be deleted, without impacting the verifiability of other resources signed by a contract.": I think that if you delete facts, then the signature of the DTC becomes invalid.
- Section 5.2: "Performance & Scalability. If a sufficiently large number of resources are signed per-contract, the per-resource storage communication overhead falls below 1 kB rela- tively quickly.": what do you mean by "sufficiently large"? How can you be sure that "the per-resource storage communication overhead falls below 1 kB"?

---

> ### Author Rebuttal · Authors · 2021-01-29
>
> To address the reviewer's valuable comments, we will clarify all mentioned ambiguities concerning bootstrapping collaborations, the signature creation process, etc., in the following. Furthermore, we briefly discuss the security of the DTC generation protocol.
>
> ## System Design & Realization
>
> ### Knowing RIDs in advance
>
> The stated issue corresponds to the bootstrapping of data sharing, i.e., how a collaborator decides which data to request.
>
> ### Protocol security analysis
>
> Due to the page limit, we cannot add a formal security proof of ReShare. Given that we only simplistically incorporate building blocks, ReShare is secure by design. The canonicalization of LD resources to create checksums is an exception. However, it does not cause vulnerabilities, as DTCs will not be accepted if the checksums cannot be verified. In general, attacks on the generation handshake are detectable through the verification mechanism, and the practical value of business relations is expected to outweigh potential benefits from exploiting the DTC generation mechanism.
>
> ### Signature inputs
>
> Both signatures cover the identities, checksums, the timestamp and optional custom content, and exclude the signatures themselves. We will make sure to clarify this part.
>
> ### Checksum computation overhead
>
> ReShare supports a pre-generation of fact checksums. Alternatively, rate limiting and access control could be used to mitigate such DoS attacks. We will make sure to discuss this issue in the final version of the paper.
>
> ### Maintaining contractIRIs
>
> Contract IRIs are only used as convenient identifiers for LD compatibility. The IRI does not need to be resolvable, and even if IRIs would not be maintained indefinitely, e.g., if the server assigns the same IRI to multiple contracts, it does not impair the trustworthiness and reliability provided by DTCs.
>
> ### Inconsistent terminology & missing timestamp (Fig. 2)
>
> Thank you for pointing out these inaccuracies. We will fix them for the final version of the paper.
>
> ## Example Use Case
>
> ### Weak use case example, no scalability required
>
> While this concern might be valid at the moment, tomorrow's industrial manufacturing settings significantly increase the amount of produced data, thus add increased scalability requirements [2].
> Furthermore, management overhead for aggregating and bundling type data for archival could be reduced by directly archiving relevant data together with DTCs.
>
> ### Definition: "massive amounts of production and design data"
>
> We again refer to related work [2] for a discussion of tomorrow's needs.
>
> ### Example readability
>
> We will extend our short design overview of ReShare in the introduction to Section 4 to improve the readability of the presented example.
>
> ## Evaluation
>
> ### Evaluation of (a) large numbers of facts, and (b) with facts changing frequently
>
> We agree that a per-contract evaluation of time needed and communication bytes for high numbers of facts per contract (a) would be interesting to evaluate feasibility. We expect that the server module could not be run on restricted devices in such a scenario. However, it is questionable whether a single device could maintain such an amount of facts, thus making a capable centralized resource management system necessary.
>
> As we do not differentiate between modified and new facts due to the immutable data scheme with revision identifiers in IRIs, (b) would arguably not provide new insights.
>
> ### Evaluation datasets
>
> While it would generally have been possible to analyze the impact of all described parameter combinations in a single experiment based on a single dataset, the results would have been hard to visualize and discuss in the format of a paper. We therefore split the evaluation into two orthogonal discussions of parameter combinations as visualized in Fig. 3 and Fig. 4 and discussed in Section 5.1. We will improve the description of our evaluation setup to clarify this.
>
> ### RTT estimate
>
> In the handshake, both parties alternate between receiving and sending messages.
> Therefore, if the effect of an increased delay is $n$ times the amount of delay increase, this hints at $n$ messages, equivalent to $\frac{1}{2} n$ RTTs.
>
> ### Per-resource storage & communication overhead
>
> Fig. 3 shows that this happens for every protocol mode and number of facts per contract $\geq 30$. By communication overhead, we here only mean the overhead introduced by ReShare, not by transmitting or storing the underlying data.
>
> [1] http://doi.org/10.1145/3338499.3357357
>
> [2] http://dx.doi.org/10125/60162

---

> > ### Comment · AnonReviewer3 · 2021-02-09
> > **Answer to Rebuttal**
> >
> > Thank you for comments and explanations. I still think that the use case and the evaluation are two weak points of your article.

---

### Official Review · AnonReviewer5 · 2021-01-13
**Data Reliability and Trustworthiness through Digital Transmission Contracts**

**Rating:** 1
**Confidence:** 4
**Impact:** 4
**Design And Technical Quality:** 4

**Review:**

The paper proposes a new signature model and handshake for JSON-LD data. The model uses well-known Web interactions to conduct a message exchange between the data source and sink, using cryptographic signatures to prove the correctness of the transferred data.

The identified issue is relevant, and the authors correctly state that significant further research is required in the domain. The promoted contract model combines best practices from the Semantic Web with established trust mechanisms. The core contribution however, the contract scheme and the handshake, do not consider the huge amount of similar works from the (Semantic Web) community. Furthermore, the evaluation gives only limited insights about the general feasibility and whether the original claims are fulfilled. Together, these two concerns make it very hard to estimate if the proposed method provides an advantage over the state of the art.

### Minor Remarks
* The authors might want to think about using a different example. The recent events about Boing jets and FAA’s involvement create certain associations in the reader’s minds, which are not helping the intended message.
* FÄA --> FAA?


## After Rebuttal
The authors have provided a detailed response to many of my raised concerns. I want to thank the authors for their effort to clarify these points. Still, I am not yet completely convinced that the main design decision, to add the signatures to the signed content, is a good practice. However, as the authors will provide more explanations in the CR, the overall approach can be discussed at the conference. I have updated my score accordingly.

### Further remark:
Even though I highly appreciate Widocu to document ontologies, I would ask the authors to extend the provided explanations on the website. The proposed ontology creates value not by the definition and annotation of the classes - quite hard to understand the background only by the comments and labels - but the interaction sequence. A few examples, documentations of each step and what needs to be done when, and which class/property to use, would be helpful. This is (currently) completely missing, giving every developer a really hard time to understand/implement the concepts with no further material.

**Anonymity:**

Yes, I would like my review to remain anonymous.

**Reuse And Availability:**

3: Medium

**Strong Points:**

* The topic of distributed trust in a data ecosystem is critical and one of the core challenges for real data-driven business cases. The identified goals are correct, and there is enough ‘space’ for innovative ideas like the ones presented in the paper.

* The paper is well written and easy to follow. The motivation for the conducted research is clear and understandable, and the relevance for a combined model of trust and interoperability is given.


**Subreviewer:**

I submitted this review.

**Weak Points:**

* As stated, I have severe concerns regarding the claimed contributions of the paper. It seems like the proposed approach is independent of JSON-LD but rather bound to JSON itself. If this is correct, the outlined model is in direct competition to well established standards, most prominently JSON Web Tokens (JWT) and JSON Web Signature (JWS). Even more important, the work of the W3C Digital Verification Community Group [1] is not mentioned at all. It is not sufficiently pointed out why the ReShare model is better than the already existing technologies, or why it can have a higher impact.

* The related work section is missing many important contributions. In particular, the Linked Data Platform [2] interactions seem like a proper inspiration for the creation/transfer of LD resources. ODRL [3] might provide a suitable scheme for the contract model.

* In the example of Fig. 2, the signatures are part of the JSON-LD statement itself. However, by adding the signature, the resulting graph gets a different signature, thereby directly falsifying itself. As the text does not mention anything else (please correct me if I missed it), the reader needs to assume that the signatures appear together with the signed data. This problem can certainly be solved somehow but needs to be stated and discussed.

* I was not able to find the ReShare ontology. The only thing I found was an ontology by Boni et al. [4], which most probably is something completely different.

* The added value of the handshake is not clear. As far as the underlying challenge is explained, a single signature by the sender should be sufficient. If the sender needs the confirmation of the receiver, a logging of the HTTPS response might even be sufficient.

* I was not able to understand why non-HTTPS methods are useful. The necessary computation steps are already quite resource-intensive, making it impossible for IoT devices (which is by its own not a problem). JSON-LD itself is too heavy for most restricted devices. However, if some kind of edge controller is required anyway, the overhead through HTTPS shouldn't be an issue anymore.

* The qualitative evaluation provides not many insights. It is hard to see why the presented results support the feasibility of the claimed contributions.

[1] https://www.w3.org/TR/vc-data-model/

[2] http://www.w3.org/TR/ldp/

[3] https://www.w3.org/TR/odrl-model/

[4] Boni, Mohammad & Abuomar, Osama & King, Roger. (2014). ReShare: An Operational Ontology Framework for Research Modeling, Combining and Sharing. Proceedings - 2014 International Conference on Computational Science and Computational Intelligence, CSCI 2014. 1. 327-333. 10.1109/CSCI.2014.63.

---

> ### Author Rebuttal · Authors · 2021-01-29
>
> We are grateful for all the comments given and the criticism outlined. In the following, we will clarify the benefit of receiver signatures, elaborate on the creation of signatures, and provide the missing reference to the ReShare ontology.
>
> ## System Design & Realization
>
> > The added value of the handshake is not clear. As far as the underlying challenge is explained, a single signature by the sender should be sufficient. If the sender needs the confirmation of the receiver, a logging of the HTTPS response might even be sufficient.
>
> Assume a use case scenario where the sender is responsible for guaranteeing it supplied the receiver with certain data. Without a signature by the receiver, any statement made about supplying said data can easily be doubted, as the sender can arbitrarily forge signatures. Because TLS uses symmetric encryption, the sender could also forge a receiver response. Thus, neither a single sender signature, nor logging the HTTPS response would be sufficient to ensure data immutability. Consequentially, we require a full handshake to provide the needed guarantees (especially for settings with mutually distrustful stakeholders).
>
> > Doesn't adding a signature to an already signed DTC graph falsify the previously existing signature? How is this addressed?
>
> The signatures present in a DTC are excluded from signature creation: When creating the sender's signature, no signature exists yet, and to create the receiver's signature, the sender's signature is excluded from the input, as we illustrate in Fig. 1. Therefore, to verify the signatures, the signatures themselves are removed from the input. We will make sure to add clarifying statements to the respective sections in a revised version of the paper.
>
> > I was not able to find the ReShare ontology.
>
> We apologize for not providing a reference, which we will add to the revised version. The ontology is publicly available [1].
>
> > I was not able to understand why non-HTTPS methods are useful, as the system cannot be deployed to IoT devices anyway.
>
> The use case-specific needs w.r.t. security and deployability will show whether any added overhead is justified. However, as mentioned by the reviewer, IoT gateways are a promising approach to easily introduce the benefits of ReShare into existing IoT systems.
>
> ## Evaluation
>
> > The qualitative evaluation provides not many insights. It is hard to see why the presented results support the feasibility of the claimed contributions.
>
> By discussing the fulfillment of our design goals, which we identified as essential requirements for use case scenarios beforehand, we distinguish our system from existing approaches, and show its benefits and limitations, especially w.r.t. performance, scalability, trustworthiness, and reliability. We will gladly address any additional feedback if it stresses that the design goals' fulfillment is not sufficiently discussed.
>
> ## Related Work
>
> > The outlined model is in direct competition to JSON Web Tokens (JWT) and JSON Web Signature (JWS), and the work of the W3C Digital Verification Community Group.
>
> None of the mentioned technologies use bilateral signatures, whose importance we stressed before. The contribution of ReShare is not the mechanism of encapsulating data signatures in a JSON format, but rather the novel concept of bilateral signatures to provide data immutability, and thereby reliability.
>
> > [T]he Linked Data Platform interactions seem like a proper inspiration for the creation/transfer of LD resources.
>
> Linked Data Platforms indeed constitute a relevant application scenario for ReShare. However, as stated in our design goals, we explicitly strive to maintain compatibility with any resource-oriented application protocol.
>
> > ODRL might provide a suitable scheme for the contract model.
>
> ODRL specifies a data usage policy language, which we consider interesting related work orthogonal to the trustworthiness and reliability (through integrity, authenticity, non-repudiation and immutability) provided by ReShare.
>
> ## Use Case Example
>
> > The authors might want to think about using a different example due to recent events.
>
> We believe that due to the well-known and long-established reliability requirements in the aerospace industry, our selected example is a good fit. Even more, recent incidents even show the practical relevance of data reliability. Thus, we believe that readers can easily understand the outlined need for data reliability (in supply chains).
>
> [1] ReShare Ontology v0.1, http://i5.pages.rwth-aachen.de/factdag/reshare-ontology/0.1/

---

### Official Review · AnonReviewer2 · 2021-01-15
**Data Reliability and Trustworthiness through Digital Transmission Contracts**

**Rating:** 2
**Confidence:** 3
**Impact:** 3
**Design And Technical Quality:** 4

**Review:**

Authors of this paper focused on the challenges of current decision-making processes which heavily rely on data driven solutions that demand high reliability and trustworthiness of data.

In this regard, authors identify the need for digitally-verifiable data immutability as an enabler for reliability and trustworthiness, and, introduce ReShare.

ReShare is a concept of digital transmission contracts that allows users to verify and prove data immutability, enabling both digital accountability non-repudiation, through the use of integrity, authenticity and signatures verification. Additionally, the proposed approach should ensure scalability and performance, especially, when compared to state-of-the-art promising approaches such as those based in distributed ledger technology which in turn introduce substantial overhead hindering its applicability in various domains.

In order to validate the applicability and performance of ReShare, authors introduce a use case from the Aerospace Engineering domain which has very strict safety demands and therefore justifies the need for achieving data immutability as well as accountability and non-repudiation.

In general, this paper is interesting to read, interesting for the conference and a good piece of work. it addresses important challenges encountered in today’s data (sharing) driven ecosystems where decision making heavily relies on it and its authenticity, reliability, and trustworthiness are of utmost relevance, especially in semi-automated decision-making processes.
The paper presents original work, it’s nicely presented, clearly organized and easy to read.


**Anonymity:**

Yes, I would like my review to remain anonymous.

**Reuse And Availability:**

4: High

**Strong Points:**

Authors present original work and compare their work to state-of-the-art approaches clearly showing how their approach advances the state-of-the-art.

The topic in general is relevant to the conference and addresses current challenges in data driven decision making processes. Authors justify their approach by introducing a use case with high safety demands.

The paper present an initial evaluation (qualitative and quantitative) with regards to performance and also in the sense of properties achieved when compared to current state of the art work.

The paper is well organized and well presented.


**Subreviewer:**

I submitted this review.

**Weak Points:**

In general the paper presents work of some maturity and couple of weak points have been identified by the reviewer which could be improved.

- Design goals-
Although, it is not so complicated to interpret what authors referred to when establishing the design goals, there is a room for improvement especially in the terminology used.
*G1. LD Signatures*  - it seems to me that the goal should have been defined in terms of what is needed to be achieved, meaning non-repudiation, integrity and authenticity, which in turn could be achieve using LD signatures and not the other way around. In the way it is presented it is more a solution than a goal.

*G3. Interoperability and Usability* - while the explanation of it makes sense, the use of both terms is a bit misleading.
The use of the term usability does not have a proper association in the paper,  in some contexts is used either as applicability or as utility ore even usefulness, but not as what usability seems to mean.
In case this is a misunderstanding from the reader please provide a clear definition or reference in the paper.
Similarly, the interoperability design goal seems not very related to what is presented and how it is being addressed, the (G6) payload flexibility seems more related to the interoperability. Please recheck your definitions.

- Background and related work
A comparative (summary) table showing the design goals and how they map to the state-of-the-art could highly benefit the clarity related to the needs for addressing certain aspects that have not been covered by state-of-the-art approaches.

- Use case-
The use case could be more elaborated and related to the proposed approach in terms of evaluation.
The paper does not address the complexity of the supply chain, number of data sources, etc.,

- Verification mechanism-
It is not clear how the verification mechanisms addresses the design goals specified as G3 (as stated by the authors)

- Realization-
Authors propose 4 modes for realization, it is however not clear why HTTP is considered and what is the threat model associated to this decision. It would be very interesting to have some sort of security evaluation as otherwise it is not clear how an attacker could be hindered to not jeopardize the security properties provided by the proposed approach.

- Evaluation-
One of the shortcomings from the DL technology was the applicability to IoT domains, complex supply chains and huge amounts of data, in the evaluation of ReShare it is not clear how is this addressed, how is performance achieve in such environments and the overhead is reduced when using X.509 certificates.

---

> ### Author Rebuttal · Authors · 2021-01-29
>
> We are grateful for the valuable input provided by the reviewer. In the following, we address the points raised out by the review and respond to the suggestions and criticism. We clarify the fulfillment of G3 by the verification, acknowledge the improved terminology for design goal G3, and elaborate on benefits in the context of IoT applications compared to existing approaches.
>
> ## System Design & Realization
>
> > Unclear how the verification mechanisms addresses the design goals specified as G3
>
> By design, the verification mechanism provides unambiguous verifiability through its simplicity, as this process consists only of verifying the certificates, signatures, and checksums in a DTC. To further assist this step, signature and checksum types, as well as the underlying data representation to create said checksums, are explicitly specified and modeled in DTCs. We will gladly add some additional pointers to the revised version of the paper.
>
> > Why is an HTTP mode considered?
>
> Our intention is to realize ReShare with well-established and widely available technology also w.r.t. our design goal G3.
>
> ## Design Goals
>
> > LD Signatures - the goal should have been defined in terms of what is needed [...] and not the other way around.
>
> We appreciate that the review raises this issue, which allows us to generalize our design goals further, also w.r.t. future research in this emerging area.
>
> > Recheck definitions of usability & interoperability
>
> After reviewing the concepts and their definitions, we have concluded that applicability is a more fitting term, which we will update in the revised paper. While we acknowledge that interoperability and usability are somewhat of relevance, with the current focus of the paper, we only tackle them partially, as stated by the reviewer. Thus, we will further clarify their relationship and focus in the revised version of our work.
>
> ## Evaluation
>
> > How does ReShare achieve performance / reduce overhead compared to DLTs when using X.509 certificates?
>
> The mentioned criticism is not limited to ReShare, but applies to all designs using public-key cryptography. When comparing Reshare to "traditional" immutability systems, e.g., systems based on DLT, we do not require each node to maintain a full copy of a distributed ledger, and do not burden them with computationally or communication-intense (network-wide) mechanisms, such as Proof-of-Work. Furthermore, if we expect a symmetric distribution of users among servers and clients, individual nodes (e.g., IoT devices) are not burdened with an additional overhead with a growing number of users. This scalability is in contrast to DLT, where maintaining the ledger becomes more expensive with a growing number of users.
>
> Concerning the use of X.509 certificates, we cannot give a single definite answer at this point as the overhead correlates with the chosen algorithm (mode). In general, efficient schemes, such as elliptic curve cryptography (ECC), could be used. If this explanation leaves any concerns, we kindly ask the reviewer to elaborate on them.
>
> ## Related Work
>
> > Add a comparative (summary) table: design goals vs. state-of-the-art
>
> Due to the page limit, we decided to not include such an overview. We are happy to provide an overview in the suggested form as supplementary material together with the final version. Alternatively, we are also grateful for any suggestions that allow us to shorten the paper, so that we can add the summary to the main body.
>
> ## Use Case Example
>
> > The discussed use case does not address the complexity of the supply chain, number of data sources, etc.
>
> Again, we are challenged by the page limits. Regardless, we agree that discussing the complexity of supply chains and the resulting impact would be very interesting and could provide further insights on specific requirements w.r.t. ReShare. However, such requirements also vastly differ depending on the considered use case. Consequently, there is no single use case that can fully account for this diversity. Therefore, for future work, we are already planning to look into additional real-world use cases in more detail. For now, our presented use case serves as a first, appropriate example to critically discuss the benefits of ReShare.

---

### Official Review · AnonReviewer4 · 2021-01-15
**An interesting solution to a pressing problem in sharing of trustworthy (linked) data**

**Rating:** 2
**Confidence:** 2
**Impact:** 4
**Design And Technical Quality:** 4

**Review:**

The authors address the pressing issue of sharing (linked) data in a trustworthy and reliable manner by introducing the concept of Digital Transmission Contracts and demontrate its practical applicability and scalability with a PoC called ReShare.
The paper is well written, well structured and easy to read. The motivation is well explained and the evaluation seems to be appropriate for illustrating the concept's technical feasibility within a real-world setting.


**Anonymity:**

Yes, I would like my review to remain anonymous.

**Reuse And Availability:**

4: High

**Strong Points:**

The paper is a solid piece of scientifc work. It does not just provide an important building block to improve existing deficiencies in the area of linked data, it also leverages the linked data approach to a level that makes it applicable in areas where it has not been considererd relevant before, exactly due to the fact that the problems addressed in this paper have not been solved appropriately.

Hence, this is a formally, technically and conceptually strong paper whose impact goes beyond the area of linked data.


**Subreviewer:**

I submitted this review.

**Weak Points:**

The reviewers familiarity with the topic is rather low, hence - from a technical PoV -  no weak points, just some typos detected:

p2: "That is, while coarse signatures often force users to retrieve unnecessary data [to] only to verify signatures, ..."
p7: "Resources are identified by [a] unique Resource IDs (RIDs)."
p13: "Usually, data-driven business models with high data reliability need[s] either depend upon well-trusted business partners ..."

---

> ### Author Rebuttal · Authors · 2021-01-29
>
> We would like to thank the reviewer for providing the review of our submitted paper. We appreciate the support with the raised minor issues and will address them as seen fit in a revised version of the paper. Furthermore, we believe that the provided review is a useful addition as our proposed concept of ReShare should also serve as an interesting and informative read for non-experts, especially considering the challenges of deploying such a mechanism in the wild (e.g., in real-world supply chain environments).

---

### Official Review · AnonReviewer1 · 2021-01-17
**Missing the full potential**

**Rating:** 2
**Confidence:** 4
**Impact:** 3
**Design And Technical Quality:** 4

**Review:**

The paper describes a supply chain system and intends to secure data
exchanges in this supply system. The system used in rather classic as
is the data exchanged. Instead of using XML and XML signature, XML
C14N and XML encryption to encrypt the relevant parts of the
exchanged data stream, the authors raise the issue for Linked data.

There is a consistent typo (probably from wrong latex coding) FÄA -> FAA

The state of the art is nicely done. It is good to see that someone
remembers that Jeremy Carrol has invented the named graphs to solve
the issue of signing RDF. The authors unfortunately do not
concede that Carrol's system could sign one triple if it were a named
graph, things that we see in the RDF* development. As opposed to the
authors, I do not believe that Linked data canonicalization is
solved. But one has to recognize that one article cannot solve
everything at a time.

The idea to have both parties in the supply chain signing the data is
very smart. It is also smart to only sign things that are of decisive
nature in case of dispute, which is a legal question. The intended
contract vocabulary, in turn, is undercomplex for that goal. A system
could also use a trusted third party with a root key. This isn't
explored due to raging blockchania, which is understandable.

The exploration of trust relations and what constitutes trust is done
in a classic scholar way, neglecting hat Linked data is not the
classic scholar way (yet). Which means the benefit of using Linked
Data does not benefit the trust relation because the semantics and
meaning of the data do not participate in the securing
considerations. Ok, it is easy from the solution to say that one
could also sign the ontology involved. But it wasn't done. One
shortcoming resulting from this is, that the authors give no
guidelines on which collection of triples to sign. This is left to
things in practice. But practice will show patterns. Hope is that we
will see those patterns emerging. For patterns, the authors should
look into the legal partitioning of such relations in
supply chain extranets.

Concluding, the authors make smart moves, but remain below of the
potential of the scenario they nicely depicted. But given the number
of novel ideas, the shortcomings do not constitute a reason to reject
the paper.

**Anonymity:**

Yes, I would like my review to remain anonymous.

**Reuse And Availability:**

4: High

**Strong Points:**

Bilateral signing for immutability and combination with accountability reporting

**Subreviewer:**

I submitted this review.

**Weak Points:**

Is not using the full potential of its own idea by ignoring the semantic dimension of trust. Is caught in blockchania to please reviewers.

---

> ### Author Rebuttal · Authors · 2021-01-29
>
> First of all, we want to thank the reviewer for the helpful insights and comments.
>
> In the following, we hope to clarify ambiguities pointed out by the review and respond to suggestions and criticism.
> Specifically, we argue that ReShare's complexity is sufficient for its Proof-of-Concept state and that it opens up room for future follow-up work.
> ReShare further provides the relevant feature of integrity and authenticity for LD and other Web resources through its DTC concept, improving data trustworthiness and thus constituting a relevant contribution for the LD research community.
>
> ## System Design & Realization
>
> > The idea to have both parties in the supply chain signing the data is very smart. It is also smart to only sign things that are of decisive nature in case of dispute, which is a legal question. The intended contract vocabulary, in turn, is undercomplex for that goal. A system could also use a trusted third party with a root key.
>
> To the best of our knowledge, we are the first to consider the paradigm and implications of a bilateral on-demand signature scheme, where both the sender and receiver of a transmission sign the state of transmitted data to provide data immutability. As our system presents a novel paradigm and is designed as a Proof of Concept, we leave the task of achieving sufficient complexity for real-world application to future work and standardization. Thus, these steps could also involve vocabulary support for a root key of a centrally trusted third party.
>
> > The exploration of trust relations and what constitutes trust is done in a classic scholar way, neglecting hat Linked data is not the classic scholar way (yet). Which means the benefit of using Linked Data does not benefit the trust relation because the semantics and meaning of the data do not participate in the securing considerations. Ok, it is easy from the solution to say that one could also sign the ontology involved. But it wasn't done.
>
> While LD may itself benefit the trust relationship, e.g., through the inclusion of adequate provenance information, integrity and authenticity of this data must still be established first. As we specifically target LD as a supported payload, ReShare is able to provide said integrity and authenticity. Subsequently, the combination of LD with ReShare may even provide an important building block to further promote LD adoption in practice, particularly in the Industrial Internet of Things.
>
> > [T]he authors give no guidelines on which collection of triples to sign. This is left to things in practice. But practice will show patterns. Hope is that we will see those patterns emerging. For patterns, the authors should look into the legal partitioning of such relations in supply chain extranets.
>
> While the issue raised by the reviewer is appropriate and relevant for the application of ReShare in supply chain use cases, further research concerning patterns in supply chain extranets is limited to such use cases. To avoid unnecessarily limiting the scope of use cases for ReShare, we prioritize flexibility w.r.t. data patterns. However, we acknowledge this aspect and identify the discussion of such patterns as interesting future work.
>
> ## Related Work
>
> > The authors unfortunately do not concede that Carrol's system could sign one triple if it were a named graph, things that we see in the RDF\* development. As opposed to the authors, I do not believe that Linked data canonicalization is solved.
>
> We agree with the reviewer that the area of RDF canonicalization still requires further (orthogonal) research. For our purpose of providing a Proof of Concept, the URDNA2015 algorithm, as specified in an active W3C draft community group report [1], is sufficient for preliminary application in ReShare.
>
> [1] RDF Dataset Normalization (Draft Community Group Report 21 November 2020), http://json-ld.github.io/normalization/spec/

---

> > ### Comment · AnonReviewer1 · 2021-01-30
> > **To the best of our knowledge, we are the first to consider the paradigm and implications of a bilateral on-demand signature scheme, where both the sender and receiver of a transmission sign the state of transmitted data to provide data immutability.**
> >
> > Unfortunately, some northern Italian legal scholars and business men were slightly earlier than you when they invented the bill of exchange in the 15th century. See https://en.wikipedia.org/wiki/Promissory_note in the subsection bill of exchange. You'll find the appropriate historic references in the German version. https://de.wikipedia.org/wiki/Wechsel_(Wertpapier)#Geschichte
> > Replacing a signature by some crypto-operation is a matter of evidence for the legal side and does not (for the legal side) accept the concept of signature as such. This confirms that, whether the paper gets accepted or not, you need to invest more work into the legal side of your system. The legal semantics are not sufficiently considered. They may be game changing.
> > For the rest, I'm happy with the rebuttal and maintain my accept.

---

### Decision · Program_Chairs · 2021-02-23

**Decision:**

Accept

**Comment:**

The reviewers agree that the work is acceptable for the ESWC research track. We invite the authors to address all the issues as promised in the rebuttal phase. Please insert all the changes in the camera-ready.